



# Measurement report:Evaluation of the TOF-ACSM-CV for PM$_{1.0}$ and PM$_{2.5}$ measurements during the RITA-2021 field campaign

Xinya Liu[1], Bas Henzing[2], Arjan Hensen[2], Jan Mulder[1], Peng Yao[1], Danielle van Dinther[2], Jerry van Bronckhorst[3], Rujin Huang[4], Ulrike Dusek*[1]

[1]Centre for Isotope Research (CIO), Energy and Sustainability Research Institute Groningen (ESRIG), University of Groningen, Groningen, 9747 AG, the Netherlands
[2]Department of Climate, Air and Sustainability, TNO, Utrecht, 3584 CB, the Netherlands.
[3]Metrohm Process Analytics, Schiedam, 3125AE, The Netherlands
[4]State Key Laboratory of Loess and Quaternary Geology (SKLLQG), Center for Excellence in Quaternary Science and Global Change, Institute of Earth Environment, Chinese Academy of Sciences, Xi'an 710061, China

*Correspondence to*: Ulrike Dusek (u.dusek@rug.nl)

## Abstract

The recently developed time of flight-aerosol chemical speciation monitor with the capture vaporizer and a PM$_{2.5}$ aerodynamic lens (TOF-ACSM-CV-PM$_{2.5}$) aims to improve the collection efficiency and chemical characterization of aerosol particles with a diameter smaller than 2.5µm. In this study, comprehensive cross-comparisons were performed between real-time online measurements and offline filter analysis with 24-hour collection time. The goal was to evaluate the capabilities of the TOF-ACSM-CV-PM$_{2.5}$ lens, as well as the accuracy of the TOF-ACSM-CV-PM$_{2.5}$. The experiments were conducted at Cabauw Experimental Site for Atmospheric Research (CESAR) during the RITA-2021 campaign. The non-refractory fine particulate matter PM$_{1.0}$ and PM$_{2.5}$ were measured by two co-located TOF-ACSM-CV-PM$_{2.5}$ by placing them behind a PM$_{2.5}$ and PM$_{1.0}$ inlet, respectively. A comparison between the ACSMs and PM$_{2.5}$ and PM$_{1.0}$ filter samples showed a much better accuracy than ±30% less given in the previous reports, with average differences less than ± 10% for all inorganic chemical species. In addition, the ACSMs were compared to a Monitoring Instrument for Aerosol and Gas (MARGA) (slope between 0.78 - 0.97 for inorganic compounds, $R^2 \geq 0.93$), and a Mobility Particle Size Spectrometer (MPSS) measuring the particle size distribution from around 10 to 800 nm (slope was around 1.00, $R^2 = 0.91$). The intercomparison of the online measurements and the comparison between the online and offline measurements indicated a low bias (< 10% for inorganic compounds) and demonstrated the high accuracy and stability of the TOF-ACSM-CV-PM$_{2.5}$ lens for the atmospheric observations of particle matters. The two ACSMs exhibited an excellent agreement, with differences less than 7%, which allowed a quantitative estimate of PM$_{1.0}$ vs PM$_{2.5}$ chemical composition. The result showed that the PM$_{1.0}$ accounted for about 70-80% of the PM$_{2.5}$ on average. The NO$_3$ mass fraction increased but the OC mass fraction decreased from PM$_{1.0}$ to PM$_{2.5}$, indicating the size-dependence on chemical composition.



# 1    Introduction

Aerosols play an important role in climate change and have been intensely studied for their effects on the global radiation balance. Direct effects include absorption and scattering of solar radiation and indirect effects refer to changes of cloud properties by aerosols acting as cloud condensation nuclei (Intergovernmental Panel on Climate Change, 2014) (Fan et al.,

2016). Furthermore, air pollution is considered the biggest environmental health threat in Europe (European Environment Agency, 2020), causing considerable morbidity and mortality (Gurjar et al., 2010; Ostro et al., 2015; Southerland et al., 2022). Approximately 7.0 million premature deaths each year are caused by long-term air pollution exposure worldwide (WHO,2019). In particular, fine aerosol particles with diameters below 2.5 µm are able to penetrate deep into the lungs, possibly causing more than 3.5 million premature deaths each year (Lelieveld et al., 2015). In the Netherlands, particulate matter is usually

dominated by secondary inorganic aerosols (SIA) due to emissions from the intensive agriculture and traffic emissions,  which has been become a serious problem to the local governments and globally (Brunekreef et al., 2009; Janssen et al., 2013; Gu et al., 2021).

Long-term monitoring of chemical composition and concentration is obviously important for controlling these emissions and improve the air quality. A lot of the measurement techniques and platforms have been developed and studied over the decades

with the aim of the long-term measurements of aerosols. The aerosol chemical species monitor (ACSM) has been developed for monitoring aerosol chemical composition, based on the aerosol mass spectrometer (AMS) (Ng et al., 2011). Compared to the AMS, the ACSM is portable, economical and relatively easy to operate.

ACSMs have been widely applied since 2011 and are continuously being improved (Wang et al., 2019). The initial design of ACSM, which has been used in most reported papers to date, was equipped with an aerodynamic lens, a standard hot vaporizer

and a lower cost residual gas analyser (RGA) quadrupole mass spectrometer (Q-ACSM) detector (Wang et al., 2019). Then the Time-of-Flight ACSM (TOF-ACSM) was developed (Fröhlich et al., 2013), which has faster response time and higher sensitivity and is used increasingly in recent years. The recent equipment of the ACSM with a capture vaporizer (Jayne and Worsnop, 2016) and a $PM_{2.5}$ lens (Xu et al., 2017) has opened avenues for quantitative study of the chemical composition of $PM_{2.5}$. One potential application is monitoring the chemical differences between $PM_{1.0}$ and $PM_{2.5}$, which have been studied

intensively for air quality monitoring. However, most previous studies comparing $PM_{1.0}$ and $PM_{2.5}$ have some limitations: most often, the comparisons were based on offline filter samples, which lack high temporal resolution (Sarti et al., 2015; Zhang et al., 2018; Giugliano et al., 2005; Vecchi et al., 2004; Perrone et al., 2013). For online approaches, the measurements usually switched between $PM_{1.0}$ and $PM_{2.5}$ by changing the size cut-off of the sampler inlet, making the comparison not exactly synchronous (Zheng et al., 2020; Sun et al., 2020) or observations of $PM_{1.0}$ and $PM_{2.5}$ were based on different instruments, thus

observed differences might result from different measurement approaches (Rodríguez et al., 2008; Budisulistiorini et al., 2014). However, to use the ACSM for such intercomparing studies requires higher accuracy than the ± 30% cited by the manufacturer, based on the standard setup with $PM_{1.0}$ lens. In this study we want to investigate if the introduction of the capture vaporizer (CV) and the $PM_{2.5}$ lens sufficiently improved the accuracy and precision of the TOF-ACSM for quantitative $PM_{2.5}$ monitoring.



In the ACSM instrument particles are converged into a narrow beam in the aerodynamic lens and then collide with the
vaporizer. The generated vapour is detected with a time of flight or quadrupole mass spectrometer after ionization (Ng et al.,
2011). The ACSM equipped with the standard vaporizer (SV) have been most frequently used to date and has been evaluated
in several previous studies (Zhang et al., 2017; Pieber et al., 2016; Xu et al., 2017; Canagaratna et al., 2015; Xu et al., 2017).The
SV has an inverted cone structure with a porous tungsten surface, which causes particle bounce and therefore reduced collection
efficiency. To reduce the particle bounce associated with the SV, the "capture vaporizer" (CV) was introduced in 2016 (Jayne
and Worsnop, 2016). The CV is made of solid molybdenum and constructed with a narrow entry "cage" and an internal
structure that facilitates repeated internal bounces. This increases the residence time of the particles in contact with the thermal
evaporator surfaces and therefore reduces the proportion of particles that bounce without evaporating (Hu et al., 2017). It has
been reported the CV can achieve a collection efficiency (CE) of 1 for ambient aerosols (Hu et al., 2017), whereas the CE of
the SV is only typically ∼0.5 for ambient aerosols, and even lower for laboratory aerosols (Matthew et al., 2008; Robinson et
al., 2017; Liao et al., 2017; Middlebrook et al., 2012).

Further, the ACSM initially measured particles with aerodynamic diameters below 1.0 µm, due to the low transmission
efficiency of the aerodynamic lens for the larger particles (Xu et al., 2017). The high-pressure aerodynamic lens (HPL) was
developed and used for the transmission of larger particles. However, the HPL requires very high precision in the machining,
which makes it difficult to reproduce consistently during manufacture (Williams et al., 2013; Xu et al., 2017). To overcome
these limitations, Peck et al., (2016) built a new intermediate pressure lens (IPL) (3.8 Torr) and it clearly improved the
transmission efficiency of particles from 1 µm to 2.5 µm (Xu et al., 2017; Peck et al., 2016). For a typical ambient $PM_{2.5}$ size
distribution, the $PM_{2.5}$ aerodynamic lens system on a Q-AMS detected a higher percentage of non-refractory mass compared
to the old $PM_{1.0}$ aerodynamic lens system. Specifically, the new system detected 89% of the non-refractory mass, while the old
system only detected 65% (Xu et al., 2017). A few articles reported the application of this new $PM_{2.5}$ inlet system (Zhang et
al., 2017), but a comprehensive assessment is still missing.

In this study, two identically configured and collocated TOF-ACSM-CV with both $PM_{2.5}$ aerodynamic lens were deployed to
measure the NR-$PM_{1.0}$ and NR-$PM_{2.5}$ during "Ruisdael Land-Atmosphere Interactions Intensive Trace-gas and Aerosol
measurement campaign" (RITA-2021) at the CESAR site in the Netherlands. Other online instruments such as the Monitoring
Instrument for Aerosol and Gas (MARGA), and Multi-Angle Absorption Photometer (MAAP), as well as a Mobility Particle
Size Spectrometer (MPSS) were applied for auxiliary measurements. Offline filters were collected and analysed to evaluate
the TOF-ACSM-CV-$PM_{2.5}$ lens. Cross-comparisons between online and online, online and offline were conducted to
investigate the capacity of TOF-ACSM-CV-$PM_{2.5}$ in long-term field measurements, and to give insights into the local chemical
composition of the NR-$PM_{1.0}$ and NR-$PM_{2.5}$.



## 2 Methods

### 2.1 Site and campaign description

A series of comprehensive aerosol in-situ measurements were performed during the 2021 RITA (Ruisdael land-atmosphere interactions Intensive Trace-gas and Aerosol) campaign, at the CESAR site in the Netherlands (51.97∘ N, 4.93∘ E). The Cabauw Experimental Site for Atmospheric Research (CESAR) is part of the ACTRIS[1] (Aerosol, Clouds and Trace Gases Infra-Structure) and ICOS[2] (Integrated Carbon Observation System) and is one of the core observation sites for the Ruisdael Observatory[3] in European and global climate networks (Knoop et al., 2021). The site is located between the northeast of Rotterdam and the southwest of Utrecht and the air masses are mostly influenced by the continental and marine environments depending on the wind direction. Previous studies showed that clean air masses are often received from the North Sea or Scandinavia. In contrast, polluted air masses generally originate from Southern Europe (Mamali et al., 2018). Continuous observations of aerosol physiochemical properties were conducted during the RITA-2021 campaign from May 11th to May 24th, and from Sep16th to Oct 12th 2021, additional measurements such as meteorological data from the 213 m high mast of Cabauw tower at 10-minute time resolution were available via the KNMI Data Platform[4].

### 2.2 Aerosol physical properties

Ground-based observations of aerosol physical properties were performed in the Cabauw main building using an inlet that samples air from 4.5 meter above the ground through the roof. Every inlet consisted of 3 parts (a) a $PM_{10}$ size selector, (b) a wide diameter Nafion drying system to dry the ambient aerosol to below 40% RH, (c) a manifold to split the aerosol flow to the multiple instruments. The inlet systems were vertically oriented to avoid deposition losses. To minimize diffusional losses all tubing was stainless steel. The measurements used in this study included: (1) A multi-angle absorption photometer (MAAP model 5012, Thermo Fisher Scientific Inc, Franklin, MA) measuring at a single nominal wavelength of 637 nm with a 5-minute time resolution to quantify the aerosol absorption coefficient (Petzold and Schönlinner, 2004). The mass concentration of equivalent Black Carbon (eBC) was calculated based on the optical absorbance at two different angles using a constant scattering cross section value (6.6 m²/g) (2) A Mobility Particle Size Spectrometer (MPSS, TROPOS) consisting of a bipolar particle charger (KR-85), a differential mobility analyzer (DMA, model Vienna-DMA medium), and a condensation particle counter (CPC 3750 TSI). Particle number size distributions in the diameter range between approximately 8 and 800 nm were recorded with a time resolution of 5 min. The inversion of the raw data was performed by a custom evaluation software (DMPS-Inversion-2.13.exe), described in (Wiedensohler et al., 2012) .

---

[1] http://actris.net/ (last access: 20 July 2022)

[2] https://www.icos-cp.eu/ (last access: 20 July 2022)

[3] https://ruisdael-observatory.nl/ (last access: 20 July 2022)

[4] https://dataplatform.knmi.nl(last access: 20 July 2022)





## 2.3    Aerosol chemical composition measurement

### 2.3.1    Online measurements by TOF-ACSM and MARGA

The nonrefractory (NR) chemical compositions of $PM_{1.0}$ and $PM_{2.5}$ were measured continuously during the RITA-2021
campaign with a time resolution of 5 min, including ammonium ($NH_4^+$), nitrate ($NO_3^-$), sulphate ($SO_4^{2-}$), chloride ($Cl^-$), and
organics (OA), using two TOF-ACSMs (Aerodyne Research Inc., Billerica, MA) (Fröhlich et al., 2013), both equipped with
CV and $PM_{2.5}$ aerodynamic lens (Xu et al., 2017). The two TOF-ACSMs were installed side by side in a trailer which was
around 200m away from the above-mentioned main measurement site. Teflon Coated Aluminium cyclones (URG 2000-30ED)
were installed at the head of the inlet with a downward entry direction to avoid external effects such as rain. Flow rates of 2.3
L min⁻¹ and 5.0 L min⁻¹ were applied to select $PM_{2.5}$ and $PM_{1.0}$, respectively. Then a multi-tube Nafion dryer (Perma Pure,
New Jersey) was used to dry the particles. It should be emphasized that the size selection occurred at ambient conditions, thus
the upper limit of the dry particle size depends on humidity. The working principle of the TOF-ACSM is based on the Aerodyne
aerosol mass spectrometer (AMS), and can be briefly described as follows: the particles are focused and drawn into the
instrument through an aerodynamic lens, then the non-refractory constituents are evaporated rapidly by the capture vaporizer
(T = 600 °C) and subsequently ionized by electron impact. The ions are identified by their mass to charge ratio in the time of
flight mass spectrometer. In the end, the electrical signal is converted to a digital signal by the signal detector and recorded
(Fröhlich et al., 2013). Several calibrations need to be performed regularly to ensure the accuracy of the instruments, including
the lens calibration, flow rate calibration and the tuning of the heater bias (HB) voltage, as well as the Ionization efficiency
(IE) and the relative ionization efficiency (RIE) calibrations. The standard procedure of the calibration details can be found in
previous publications (Fröhlich et al., 2013; Canagaratna et al., 2007). The IE and RIE calibration were performed before the
RITA campaign and the parameters used in this paper are summarized in Table 1. The data analysis was produced by Tofware
(v3.2.4, Tofwerk AG, Thun, Switzerland) based on the Igor Pro 8.

**Table 1: The setup details for two of the TOF-ACSMs and the corresponding IE and RIE calibration values for each species used in
this study**



| | TOF-ACSM PM$_{1.0}$ | TOF-ACSM PM$_{2.5}$ |
|---|---|---|
| Sampling inlet setup | URG 2000-30ED,flow rate 5.0 LPM | URG 2000-30ED,flow rate 2.3 LPM |
| Sampling dryer setup | Nafion dryer (Perma Pure, New Jersdy) connected to ARI sample line Flow controller (S/N fcb-03 and Greater) | Nafion dryer (Perma Pure, New Jersdy) connected to ARI sample line Flow controller (S/N fcb-03 and Greater) |
| Vaporizer | Capture | Capture |
| IE NO$_3$(pg s$^{-1}$) | 114.50 | 258.20 |
| RIE NH$_4$ | 3.25 | 3.51 |
| RIE SO$_4$ | 1.26 | 1.33 |
| RIE Org | 1.40 | 1.40 |
| RIE Chl | 1.30 | 1.30 |
| AB (E+5 ions s$^{-1}$) | 2.26 | 4.55 |
| Flow (cm3 s$^{-1}$) | 1.33 | 1.46 |

The Monitor for AeRosols and Gases in ambient Air (MARGA 2060, Metrohm Applikon B.V., the Netherlands) was used during the September part of the campaign to measure the water-soluble inorganic components based on ion chromatography (IC), including hydrochloric acid (HCl), Nitric acid (HNO$_3$), Nitrous acid (HONO), Sulfur dioxide (SO$_2$), Ammonia (NH$_3$) in

the gas phase, and Chloride (Cl$^-$), Nitrate (NO$_3$$^-$), Sulfate (SO$_4$$^{2-}$), Ammonium (NH$_4$$^+$), Potassium (K$^+$), Calcium (Ca$^{2+}$), Magnesium (Mg$^{2+}$) in the aerosol phase. A "MARGA-sizer" introduced by ten Brink (2007, 2009) was used to control the size of the particles (e.g. PM$_{1.0}$, PM$_{2.5}$ or PM$_{10}$) entering the instrument. We applied the PM$_{1.0}$ sizer in the first stage (from September 5$^{th}$ to September 30$^{th}$, 2021) and PM$_{2.5}$ sizer in the later stage (from October 3$^{rd}$ to October 16$^{th}$, 2021) of the campaign. The ambient air was drawn into the instrument at a constant flow rate of 16.7 LPM through a short (0.2 m) length of Teflon tubing

with an outer diameter of 25.4 mm, via a vacuum pump. Then the water-soluble gases were absorbed by a Wet Rotating Denuder (WRD) device (Wyers et al., 1993; Keuken et al., 1988) and the water-soluble aerosols were extracted in a Steam-Jet Aerosol Collector (SJAC) (Khlystov et al., 1995; Slanina et al., 2001). Eventually, the liquid of the samples was collected continuously in separate syringes and then analysed by IC at one-hour resolution. Rumsey et al provided the operational, calibration and data analysis procedures in details (Rumsey and Walker, 2016; Rumsey et al., 2014).

**2.3.2    Offline Filters measurements and analysis**

24-hour PM$_{1.0}$ and PM$_{2.5}$ filters were collected simultaneously from midnight to midnight, according to the reference method described in the European Standards (EN12341: 1998 and EN14907: 2005). The SEQ47/50 (Leckel GmbH, Germany) instrument with a sequential low-volume system (LVS) of 2.3 m$^3$ h$^{-1}$ was used for the sampling. Two polytetrafluoroethylenes (PTFE) filters (diameter 47 mm, pore size 3 μm, Millipore) and two quartz fibre filters (diameter 47 mm, Pallflex) were placed

in the samplers for a paired measurement of PM$_{1.0}$ and PM$_{2.5}$. The gravimetric mass of the PTFE filters was obtained by triple weighing before and after sampling. All the filters were protected and stored under a condition of a temperature of 20.0 ± 0.5



°C and relative humidity of 50 ± 2% during the storage and transport. The detailed information about the logistic and operational (QA/QC, weighing) procedures, as well as the data acquisition was described in Schaap et al (2010).

The $PM_{1.0}$ and $PM_{2.5}$ ion concentration analysis was done by the Ion chromatography, including the 3 inorganic anions ($NO_3^-$,
$Cl^-$, $SO_4^{2-}$) and the 5 cations ($Na^+$, $K^+$, $Mg^{2+}$, $Ca^{2+}$, $NH_4^+$). Aliquots of the filter samples (cations: 3.28 cm²; anions: 3.0 cm²) were extracted by the 2.0 mL of the 30 mM methane sulfonic acid (MSA, cations) or 2.0 mL of extra pure water (anions) for 40 minutes under ultrasonic agitation. The determination of the concentrations was performed by the ICS-1100 and AQUION instruments (both Thermo Scientific) combined with an autosampler AS-DV and ion exchange columns (cations: CS16; anions: AS22). The description of the equipment used can be found in Samek et al (2020).The organic carbon (OC) and
elemental carbon (EC) of the $PM_{1.0}$ and $PM_{2.5}$ were analyzed by a Sunset thermal-optical analyzer (TOA, Sunset Laboratory Inc.). The EUSAAR2 protocol (Cavalli et al., 2010) was used to distinguish the OC and EC, using the laser transmittance signal. The details of the operation procedure can be found in Yao (2022).

## 3 Results and Discussion

### 3.1 Intercomparison results

The comparison between online chemical composition measurements (TOF - ACSM + MAAP) and filter measurements for daily average concentration of each species is presented in section 3.1.1. In addition, the volume concentrations derived from chemical composition measurements and the particle number size distribution (PNSD) are compared in section 3.1.2 with hourly time resolution. Total online NR-$PM_{1.0}$ and NR- $PM_{2.5}$ mass concentration were calculated by adding the eBC to the total TOF-ACSM mass concentration (the sum of nitrate, sulfate, ammonium, organic, and chloride mass concentrations). The
total mass concentrations of $PM_{1.0}$ and $PM_{2.5}$ filters are also calculated by the sum of the inorganic anions ($NO_3$, $Cl$, $SO_4$), $NH_4$, OC and EC concentrations.



### 3.1.1 Comparison of online and offline measurements for PM$_{1.0}$ and PM$_{2.5}$

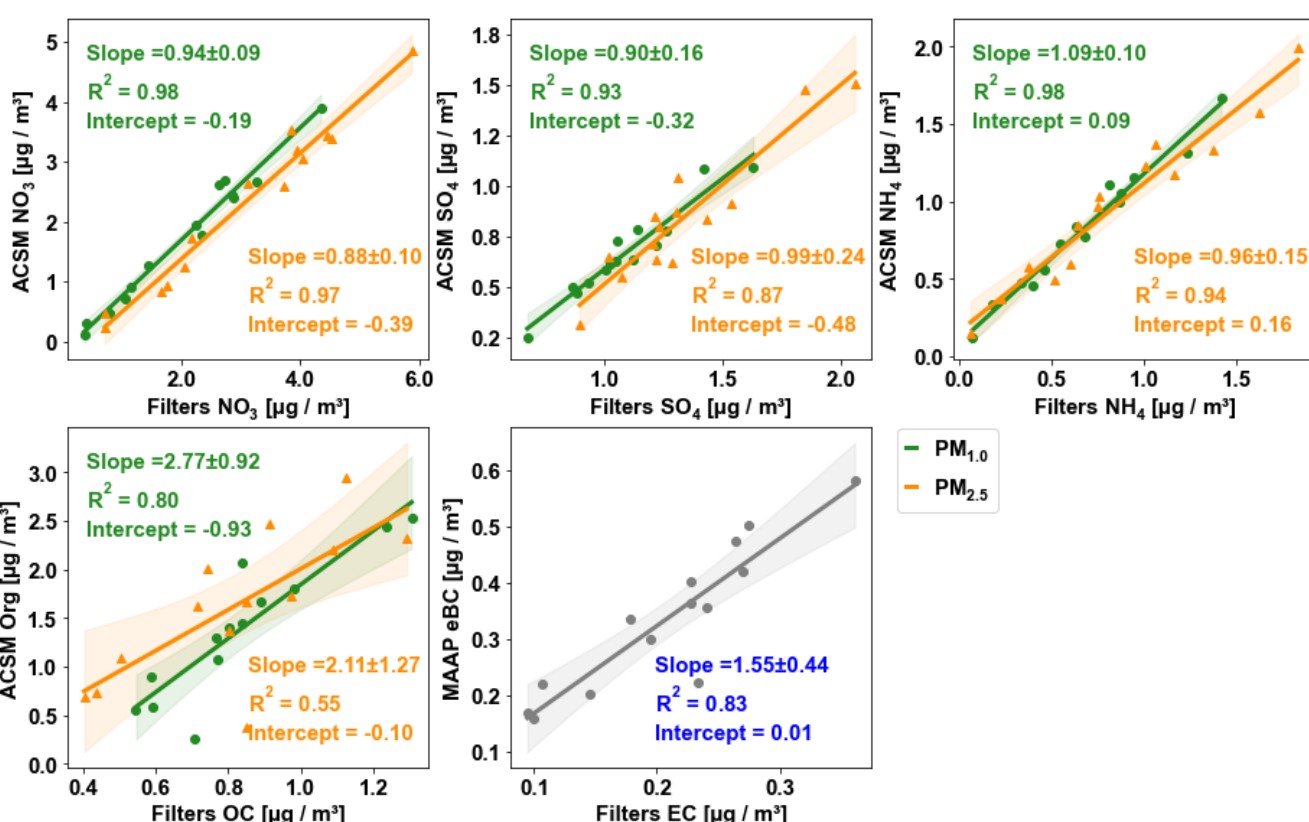

**Figure 1 Comparison of the online (ACSM and MAAP) and offline (Filters) mass concentrations of various chemical components. PM$_{1.0}$ is indicated in green and PM$_{2.5}$ in orange. The shaded area represents the 95% confidential interval of the best fit line.**

Figure 1 shows the comparison of each component between the online and offline measurements for PM$_{1.0}$ (green dots) and PM$_{2.5}$ (orange triangles) including a linear least-squares regression line. The uncertainties of the slope correspond to the standard error. Over the intensive measurement period, the daily average NO$_3$ mass concentrations measured by the TOF-ACSM-CV and by filters showed a high correlation with $R^2 = 0.98$ for PM$_{1.0}$ and $R^2 = 0.97$ for PM$_{2.5}$, and the corresponding slopes are $0.94 \pm 0.09$ and $0.88 \pm 0.10$, respectively. The results showed that NO$_3$ concentrations on the filters were slightly higher than TOF-ACSM-CV measurements. Paired t-tests were performed to investigate the significance of the difference between the online and offline measurements, and the results are shown in the Table S1 and S2. It shows a significant difference between the ACSM measured NO$_3$ and filters measured NO$_3$ (p-values are 2.38E-5 for PM$_{1.0}$ and 5.08E-7 for PM$_{2.5}$). A well-known bias between online and offline measurements of nitrate is caused by evaporation of ammonium nitrate collected on the filter, and the loss grows with increasing of temperature and decreasing of humidity (Malaguti et al., 2015; Kuokka et al., 2007; Chow et al., 2008; Pakkanen and Hillamo, 2002). Previous studies have shown that this evaporation loss from quartz





filters can exceed 80% up to complete evaporation when the temperature exceeds 25℃ (Schaap et al., 2004; Allan et al., 2003; Pandolfi et al., 2014). Therefore, the monitoring of humidity and the controlling of storage temperature have a crucial impact on the results. During the measurement in this study, the RH was 81.16% ± 14.17% and temperature was 15.94 ± 4.20℃,

which should largely prevent this evaporation loss. Consequently, we observe slightly higher concentrations on the filter samples. Likely reasons for this difference are: (i) The higher offline concentration of nitrate may also be caused by the absorption of gas-phase nitric acid (HNO3) on the filter (Chow, 1995). Bhowmik et al (2022) also observed higher nitrate concentrations on filter samples with an even lower slope of 0.49 between the online AMS and offline filters $NO_3$ measurements. (ii) For ACSM measurements, the absolute concentration of the nitrate is highly depending on the IE

calibration, which needs to be performed carefully and regularly. The calibration parameters used in this study are listed in table 1. If they are slightly biased, the ACSM concentration could be too low. But the differences are in general less than 10%, which are much better than the previous ± 30% given in the manufactory for the with a SV and $PM_{1.0}$ lens.

For sulfate, the on-line and off-line measurements also showed a high correlation, although lower than for ammonium nitrate. The slope and coefficient of determination are 0.90 ± 0.16 and $R^2$ is 0.93 for sulfate $PM_{1.0}$, and the slope is nearly 1 (0.99 ±

0.24) and the $R^2$ is 0.87 for sulfate $PM_{2.5}$. The relatively lower $R^2$ is potentially due to the low sulfate mass concentration (0.67 and 0.84 µg cm$^{-3}$ on average for $PM_{1.0}$ and $PM_{2.5}$) during the measurements. Similar to nitrate, the ACSM sulfate measurements are influenced by the IE and RIE calibrations. Apart from that, higher offline values of the sulfate may also be caused by some refractory sulfates such as potassium sulfate, calcium sulfate and sodium sulfate, which cannot be detected by TOF-ACSM (Poulain et al., 2020). Or it can also be due to the positive sampling artifacts, for example, the absorption of $SO_2$ by alkaline

particles in the filter membrane or by the reaction of gas-phase ammonia with sulfate aerosols to form ammonium sulfate or ammonium bisulfate (Nicolás et al., 2009; Nie et al., 2010). This is less likely to occur in the Netherlands as sulfate is usually completely neutralized by excess ammonia already in the ambient atmosphere.

For ammonium, the coefficients of determination were $R^2 = 0.98$ in $PM_{1.0}$ and $R^2 = 0.94$ in $PM_{2.5}$ with slopes of 1.09 ± 0.10 and 0.96 ± 0.15, respectively. As the ammonium measured by the ACSM mainly corresponds to ammonium nitrate and

ammonium sulfate, the small deviation of the online and offline data is reasonable. However, it is worth noting that the ammonia concentrations in Europe as a whole is usually sufficient to neutralize nitric and sulphuric acid (Wichink Kruit et al., 2017). In particular, an excess of ammonium (ammonium concentrations higher than those explained by the formation of inorganic ammonium salts) has been observed a lot in the Netherlands in past reports (Schlag et al., 2017). Table S4 and S5 show the molar mass concentration of cation ($NH_4$) and anions ($NO_3$ and $SO_4$) from the filter samples and ACSM

measurements. Anions are observed to be 7% higher than cation in $PM_{1.0}$ filter samples, indicated a light underestimation of $NH_4$ in Filter $PM_{1.0}$. But on the whole, the average differences between the ACSM and filter samples are less than ± 10% for all inorganic chemical species, shows a good accuracy of the ACSM with the CV and $PM_{2.5}$ lens in the field measurements.

Regarding the measurement of the organic aerosol fraction, the ACSM measures organic aerosol (OA), under the assumption that all mass, which cannot be explained by known inorganic components must be organic (Allan et al., 2004). Thus, the

quantification of the OA concentration is determined by how to interpret and assign fragmentation signals. On the other hand,



the offline measurement of the organics is normally done by thermal-optical analysis, which is usually detects only the carbon element of the organic compounds and is therefore referred to as organic carbon (OC). OC concentrations usually depend on the calculation methods and measuring protocols (Cavalli et al., 2010; Chiappini et al., 2014; Zanatta et al., 2016). As a result of the different quantification, the correlation between OM and OC is much lower than for inorganic compounds ($R^2 = 0.55$ in

$PM_{1.0}$ and $R^2 = 0.80$ in $PM_{2.5}$). Because OM also includes associated hydrogen, oxygen and other elements, OM is significantly higher than OC, indicated by a slope of from $2.21 \pm 1.27$ for $PM_{1.0}$ and $2.27 \pm 0.92$ for $PM_{2.5}$. Besides, a critical factor is called as 'the Pieber effect', which observed that the inorganic salts can thermally decompose and release carbonaceous material already present in the instrument, leading to the formation of $CO_2^+$ ($m/z$ 44) ions that are not related to the organic aerosol (Freney et al., 2019; Pieber et al., 2016). Data showed that the degree of interference was highly variable between instruments

and over time and $CO_2^+$ was overestimated by 0.4% to 10.2%. Specifically, $NH_4NO_3$ resulted in a median $CO^{2+}$ overestimate that was 3.4% higher compared to $HNO_3$. The level of interference caused by other semi-refractory nitrate salts was 2-10 times higher than that caused by $NH_4NO_3$. In contrast, $(NH_4)_2SO_4$ induced interference that was 3-10 times lower than $NH_4NO_3$. Those may have an impact on the interpretation of the OA concentration. This artifact has a more pronounced effect in aerosol environments with high inorganic salt fractions (> 50%). High inorganic concentrations in the Netherlands may have likely

aggravated this effect. Apart from this, a constant RIE of 1.4 was assumed for OA during the study based on the recommendation by Aerodyne, which can contribute to uncertainties in OA quantification, since this RIE can change for different instruments and different OA composition and concentration. Although there are some studies that attempted to convert the ACSM $f_{44}$ signal to O : C ratios and to derive OM : OC ratios, the large variability of the $f_{44}$ signal itself causes a large uncertainty in the O : C ratio (Crenn et al., 2015; Canagaratna et al., 2015; Aiken et al., 2008; Rollins et al., 2010; Poulain

et al., 2020). Thus, this approach was not attempted in this study. On average, the OM/OC ratios were $1.58 \pm 0.54$ for $PM_{1.0}$ and $1.97 \pm 0.59$ for $PM_{2.5}$ in this study, which are common ratios of OM/OC observed in the organic aerosol.

The eBC was measured online by using the MAAP with a $PM_{10}$ inlet, whereas the EC was collected on the filters using $PM_{2.5}$ inlet and then analyzed offline by the sunset analyzer. Figure 1 shows the comparison of eBC and the $PM_{2.5}$ EC with a good correlation ($R^2$ of 0.83). The slope was $1.55 \pm 0.44$, reflecting the difference in size cutoff. Secondly, the eBC measured by

the MAAP is based on a constant scattering cross section 6.6 m²/g of black carbon (Petzold et al., 2002) for converting the absorption to the mass concentration of eBC. A recent comparison between the MAAP and OC/EC analysis shows differences of 20% for an urban site and 70% for a regional site (Karanasiou et al., 2020). The 55% differences found in our studies with different size cutoffs shows a reasonable result.

To sum up, the comparison between the online and off-line measurements of the $PM_{1.0}$ and $PM_{2.5}$ showed consistent results,

especially for the SIA with slopes between 0.88 - 1.09 and the $R^2$ values greater than 0.87. The OA vs OC and eBC vs EC comparisons showed results in line with previous studies. Overall, the data were fairly accurate and reliable for further study. Especially, the configuration of TOF-ACSM-CV-$PM_{2.5}$ lens showed a high stability and accuracy. With suitable inlets it can perform both NR-$PM_{1.0}$ and NR-$PM_{2.5}$ measurements for the purpose of long-term field observation.



### 3.1.2 Comparison of chemically derived volume concentration and PNSD derived volume concentration

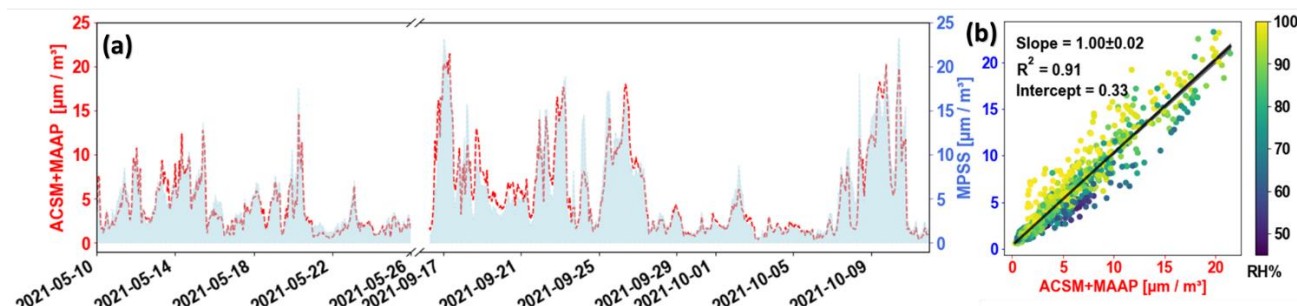


**Figure 2 (a)The time series of the ACSM and MAAP volume concentrations (red line) compared with the MPSS volume concentration (blue area). (b)The correlation of the ACSM and MAAP volume concentration with the MPSS derived volume concentration. Scatters colored by the relative humidity (%).**

The total TOF-ACSM volume concentration was also compared and validated by the particle volume concentration derived
from the PNSD. The aerosol particle size distribution with a range of around 8 - 800 nm in electromobility diameter was obtained by the MPSS during the RITA-2021 campaigns (in May and Sep) to further validate the chemical measurements. Simply put, the volume concentration from ACSM was calculated as the mass concentrations of individual species divided by the corresponding density. The density of each species using in this study was 1.75 g cm$^{-3}$ for the inorganics (Haynes, 1942), 1.2 g cm$^{-3}$ for the organics (Turpin and Lim, 2001), 1.52 g cm$^{-3}$ for chloride (Haynes, 1942), and 1.77 g cm$^{-3}$ for eBC (Park et
al., 2004; Poulain et al., 2014). The MPSS volume concentration was estimated by converting the PNSD to the particle volume distribution. The total volume concentration of the MPSS is the integral of the particle volume distribution over all the size bins. Figure 2(a) shows the time series of the volume concentrations derived from ACSM + MAAP measurements and the MPSS derived volume concentration. The agreement was good over the whole measurements period indicating a stable condition of the instrument and satisfactory quality. The correlation of volume concentrations is displayed in Figure 2(b) with
data points colored by the RH. The slope was nearly 1 ( ± 0.02) with R$^2$ = 0.91, which was comparable with previous studies (Poulain et al., 2020; Pokorná et al., 2022). However, it demonstrates that the linear correlation between the two variables is significantly influenced by relative humidity. Higher relative humidity led to a lower size cut-off diameter, resulting in a lower mass concentration measured by ACSM. As also reported in the previous studies, the aerosol hygroscopic growth has a great impact on the size cut off in terms of dry particle size (Chen et al., 2018) when the ambient RH is high. It has been pointed out
that the difference between ambient and dry cut-off size is approximately 10% and 20% for PM$_{1.0}$ and PM$_{2.5}$ in the European background, and even larger in marine or coastal stations with up to 43% and 62% for PM$_{1.0}$ and PM$_{2.5}$ (Poulain et al., 2020). The upper cut-off for the ACSM inlet is ~2.5 µm (ambient, aerodynamic) and ~ 0.8 µm (dry, electrodynamic equivalent) for the MPSS. Nevertheless, the dry, electrodynamic equivalent cut-off size of the ACSM inlet will be larger than 0.8 um. Therefore, the ACSM volume concentrations were expected to be higher and it is surprising that the agreement is so close.
However, the ACSM only measures non-refractory material, whereas the MPSS derived volume concentration also includes non-refractory material. This indicates that there is considerable contribution from non-refractory material other than BC. The




filter analysis also supports this conclusion, as seen in Figure S5, which shows approximately 21% of the PM$_{2.5}$ mass was not detected by the ion analysis. Thus, the slope of 1.00 is probably a coincidence, where missing volume from the MPSS cut-off and missing mass from the ACSM roughly cancel out. On the whole, the high $R^2$ values give confidence in the stability and

accurateness of the ACSM instrument in the long-term observations. A comparison between ACSM and MPSS volume concentrations is highly recommended as a regular quality control strategy.

### 3.2 Chemical composition of the PM$_{1.0}$ and PM$_{2.5}$.

Based on the good agreement between the online and offline measurements, ACSM accurately measured both PM$_{1.0}$ and PM$_{2.5}$ concentrations. Therefore, it is possible to further quantify the PM$_{1.0}$ vs PM$_{2.5}$ chemical composition and investigate the

differences.

### 3.2.1 Comparison of NR-PM$_{1.0}$ and NR-PM$_{2.5}$ species measured by TOF-ACSM

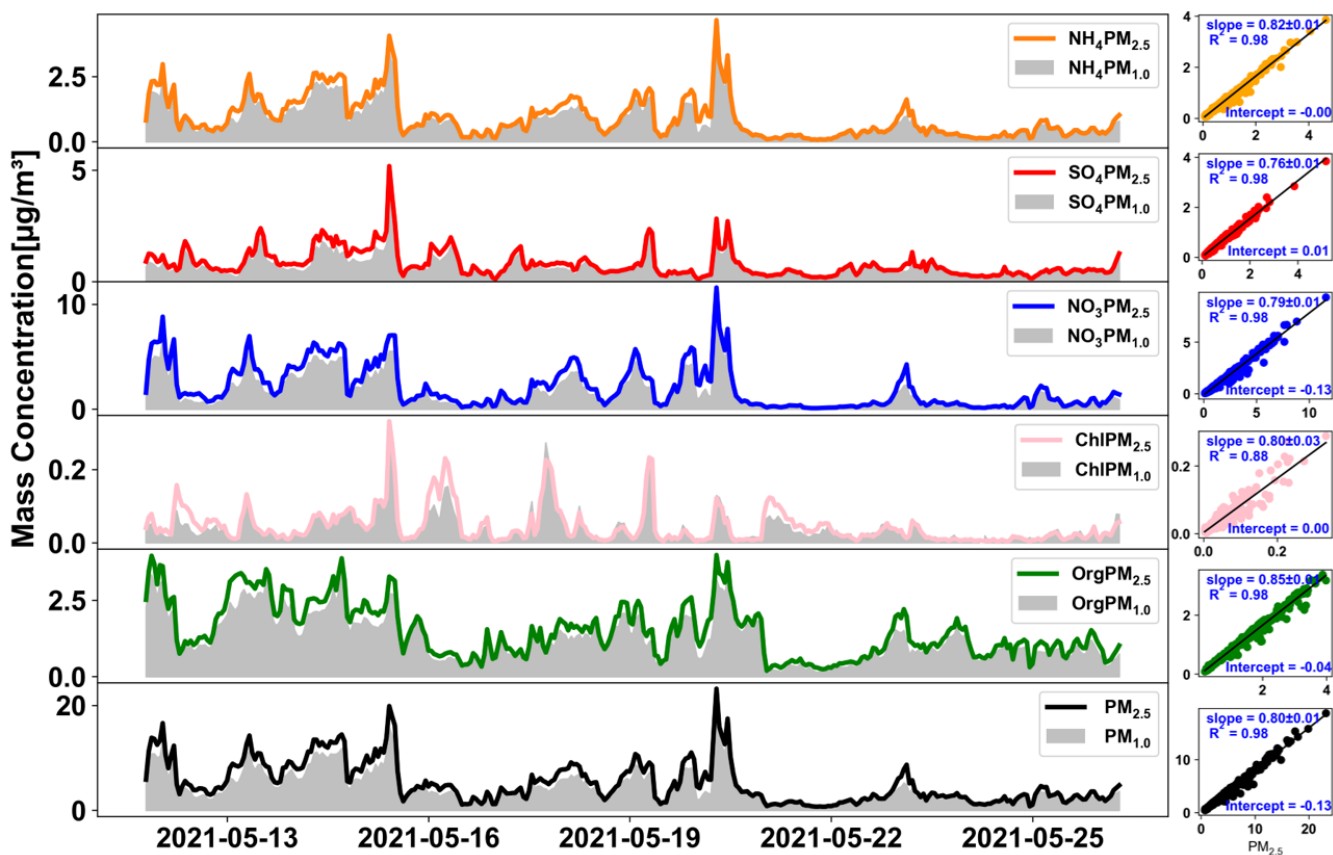

**Figure 3 Time series of the NR-PM$_{1.0}$ and NR-PM$_{2.5}$ chemical species and the total mass concentration measured by TOF-ACSM; and corresponding linear regression fitting correlations.**





As mentioned, two identically configured TOF-ACSMs with $PM_{2.5}$ aerodynamic lens were collocated and set up to measure the NR-$PM_{1.0}$ and NR-$PM_{2.5}$ during the RITA-2021 spring campaign. At the start of the campaign, both instruments were intercompared by measuring the NR-$PM_{2.5}$. The results shown in Figure S1 demonstrate good comparability, with the $R^2$ ranging from 0.91 to 1.0 and slopes ranging 0.94 to 0.99 for the SIA compounds $SO_4$, $NO_3$, and $NH_4$. The slopes were not significantly different from 1 at the 95% confidence level. For chloride, the correlation was not as good as for other species,

because ammonium chloride had a very low concentration during the whole measurement period. Therefore, it will not be discussed in the following. The correlation of $PM_{1.0}$ and $PM_{2.5}$ OA concentrations was also reasonable with a slope of $0.93 \pm 0.13$ ($R^2 = 0.80$). Overall, the two co-located TOF-ACSMs compared well and can be used to compare $PM_{1.0}$ and $PM_{2.5}$ chemical composition.

Figure 4 shows the total mass concentration time series of the NR-$PM_{1.0}$ and NR-$PM_{2.5}$ as well as the concentration of individual

chemical species, and the corresponding scatter plots with regression lines. The mass concentration of NR-$PM_{2.5}$ was on average $5.27 \pm 3.98$ µg m$^{-3}$ with a range from 4.84 µg m$^{-3}$ to 22.25 µg m$^{-3}$. This concentration was below the WHO $PM_{2.5}$ annual limit values (10 µg m$^{-3}$) (World Health Organization, 2021) and also lower than previously reported concentrations in this region of $14.4 \pm 2.1$ µg m$^{-3}$ (Schlag et al., 2016; Mensah et al., 2012; Mamali et al., 2018). The $PM_{1.0}$ and $PM_{2.5}$ mass concentrations of each species were highly correlated over the whole measurements period with $R^2 \geq 0.98$. In general, the

$PM_{1.0}$ SIA accounted for approximately 75% - 85% of the $PM_{2.5}$ SIA on average, with individual contributions ranging from $82\% \pm 1\%$ for ammonium, $79\% \pm 1\%$, for nitrate, to $76\% \pm 1\%$ for sulphate. For organics the $PM_{1.0}$ accounted for a higher fraction of $PM_{2.5}$, with around $85\% \pm 1\%$. Similar results were also found in the filter samples as displayed in Figure S2. In addition, EC-$PM_{1.0}$ accounted for $74\% \pm 14\%$ of the EC-$PM_{2.5}$. In general, the $PM_{1.0}$ mass concentration explained $80\% \pm 1\%$ of the $PM_{2.5}$ on average, and this ratio ranged from 45.21% - 94.78% throughout the campaigns. However, there was still a

substantial proportion (~21%) of unexplained mass in the $PM_{2.5}$ as shown in FigureS5.

In addition, the chemical mass fractions of $PM_{1.0}$ and $PM_{2.5}$ displayed in Figures S3-S4, revealed that there were some slight differences in the chemical composition of the $PM_{1.0}$ and $PM_{2.5}$. Figure S3 showed the average hourly mass fraction measured by the ACSM for the NR-$PM_{1.0}$ and NR-$PM_{2.5}$. The OA accounted for similar proportions namely 34.4% of NR-$PM_{1.0}$ and 33.0% of NR-$PM_{2.5}$. $NO_3$ contributed 27.8% to NR-$PM_{1.0}$ with a slight increase to 31.5% in NR-$PM_{2.5}$. Figure S4 and S5 show

the daily and the average mass fractions for $PM_{1.0}$ and $PM_{2.5}$ species from the filter samples, with a higher $NO_3$ fraction in $PM_{2.5}$ and a lower OC fraction in $PM_{1.0}$ for the whole period. Specifically, the $NO_3$ fraction increased from 38.3 % in $PM_{1.0}$ filter samples to 45.5 % in $PM_{2.5}$ filter samples, whereas the OC fraction decreased from 19.9 % to 15.1%. The difference between the ACSM OA mass fractions (similar $PM_{1.0}$ in $PM_{2.5}$) and the OC mass fraction on the filters (higher in $PM_{1.0}$ than in $PM_{2.5}$) could indicate that organic compounds were more abundant in the $PM_{1.0} - PM_{2.5}$ size range. This change in chemical

composition with particle size suggests that different types of particles may dominate in different size ranges and potentially indicating a non-internal aerosol mixing state during the measurements. The differences of OC mass fraction in $PM_{1.0}$ and in $PM_{2.5}$ also further explain the stronger correlation of ACSM OA and Filter OC in $PM_{1.0}$ compared to in $PM_{2.5}$ shown in Figure 1.



### 3.2.2   Comparison of the SIA-PM$_{1.0}$ by MARGA and SIA-PM$_{2.5}$ by TOF-ACSM

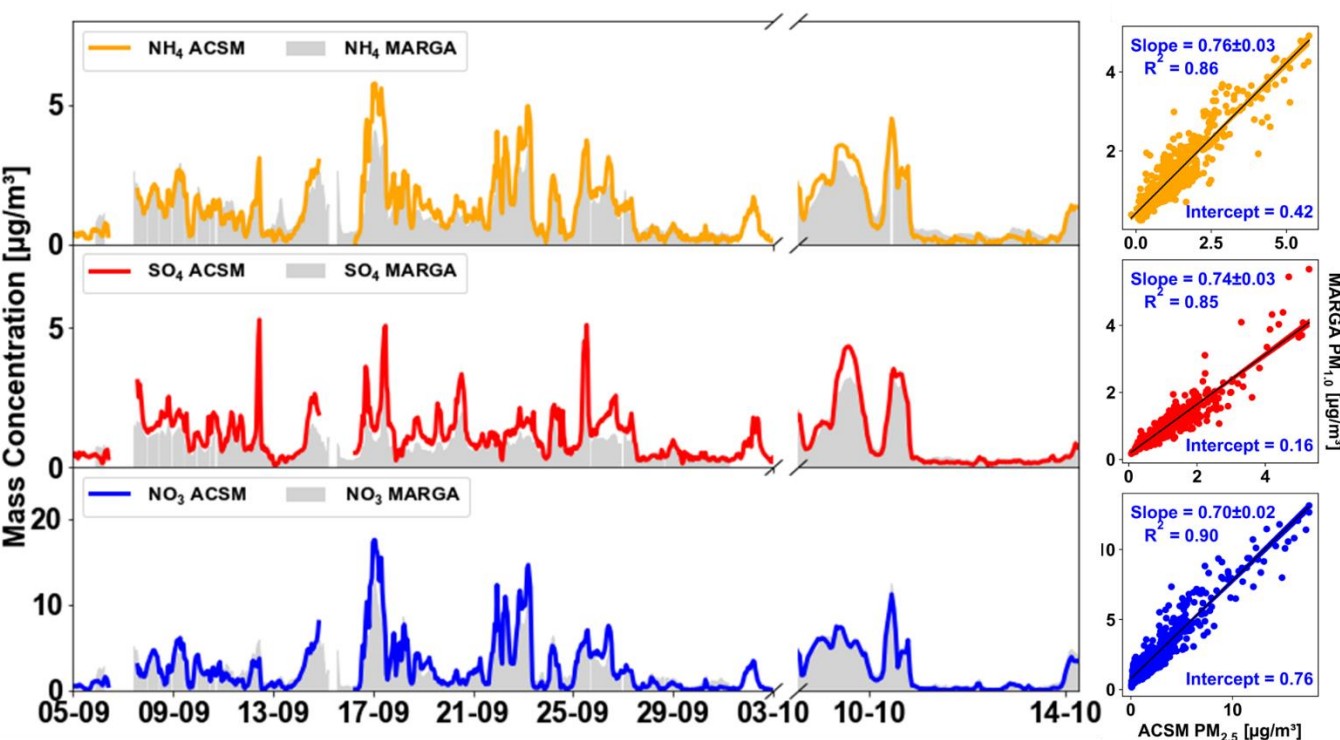


**Figure 4 Time series of TOF-ACSM measured SIA-PM$_{2.5}$ during the whole period. MARGA measured PM$_{1.0}$ from September 5$^{th}$ to September 30$^{th}$ in 2021, and PM$_{2.5}$ from October 3$^{rd}$ to October 15$^{th}$ in 2021. The corresponding linear regression fitting correlations between MARGA PM$_{1.0}$ and TOF-ACSM PM$_{2.5}$.**

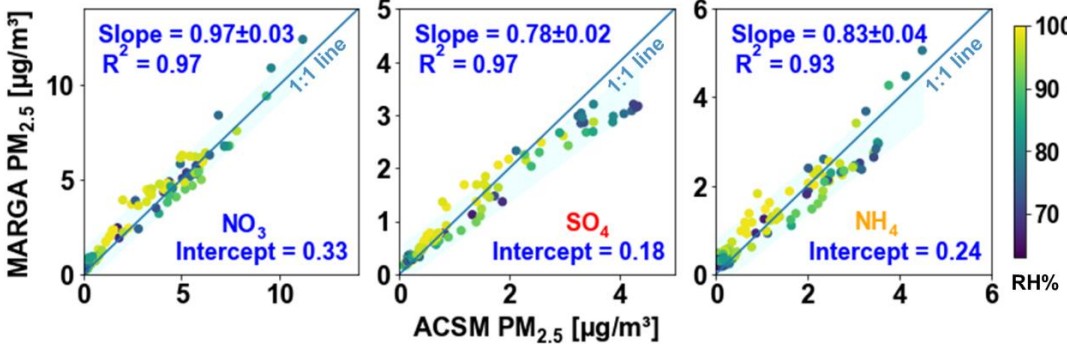

**Figure 5 The linear regression fitting correlations between MARGA PM$_{2.5}$ and TOF-ACSM PM$_{2.5}$ with points colored by the relative humidity (%). The blue shaded area represents the 95% confidence interval.**

The Comparison of the MARGA and ACSM measurements were carried out for further evaluation and validation. Figure 4 displayed the time series of the MARGA measurements and TOF-ACSM measurements. Figure 5 compares MARGA and TOF-ACSM data during time periods when both instruments measured PM$_{2.5}$. Strong correlations with R$^2$ ranging from 0.93



to 0.97 and small intercepts demonstrated a good reliability of the two methods. However, the linear regression slopes displayed some discrepancies for individual species. The $NO_3$ measured by the ACSM and MARGA showed an excellent agreement, with a difference below 3% (Slope = 0.97 ± 0.03, $R^2$ = 0.97). For the $NH_4$ and $SO_4$, the MARGA mass concentrations were lower than the ACSM mass concentrations. The slope for $NH_4$ was 0.83 ± 0.04 and for $SO_4$ it was 0.78 ± 0.02. The analysis also revealed a dependence on the RH for the correlation between the two measurements. As illustrated in Figure 5, the ACSM

tended to measure more mass than the MARGA under lower RH conditions. The hygroscopic growth of the aerosol particles at higher RH resulted in lower dry cut-off sizes and the different inlets of the MARGA and ACSM might lead to differences in the detected mass. Combined with Figure S6, it shows a slight bias between ACSM and MARGA at higher concentrations, particularly when pollution originates from the south or southeast. Since the ACSM mass also includes contributions from organic nitrates, organic sulphates, and organic ammonium salts to the observed $NH_4$, $NO_3$, and $SO_4$ concentrations, this could

also lead to higher concentrations observed by the ACSM. However, given that the validation of the TOF-ACSM against filter samples showed excellent agreement for these ions as discussed in the Section3.3.1 and listed in the Table S1-S3, the ACSM results are more likely to be closer to the true values compared with MARGA. Most previous comparisons of ACSM/AMS and MARGA showed that the MARGA gave higher concentrations when the ACSM/AMS used the $PM_{1.0}$ lens (Zhao et al., 2020; Stieger et al., 2018; Heikkinen et al., 2020). To the best of our knowledge, this is the first comparison between the $PM_{2.5}$

lens on a TOF-ACSM-CV and a MARGA. We observed that a higher concentration can be achieved by using the CV and $PM_{2.5}$ lens of the TOF-ACSM, which further verify the its capability in measuring non-refractory $PM_{2.5}$ and $PM_{1.0}$ concentrations quantitatively. Because of the very high correlation between MARGA and ACSM concentration, the slight bias between the instruments can be corrected using the regression coefficients in Figure 6. Figure 4 also displayed the linear regression correlations between the MARGA measured $PM_{1.0}$ and TOF-ACSM measured $PM_{2.5}$ inorganic chemical species

after this correction. The correlation between ACSM $PM_{2.5}$ and MARGA $PM_{1.0}$ all showed $R^2$ values greater than 0.85, and the slopes are 0.76 ± 0.03 for $NH_4$, 0.74 ± 0.03 for $SO_4$, 0.70 ± 0.02 for $NO_3$, very comparable to the slopes achieved in the spring campaign, using two different ACSMs. In summary, the local concentrations of both $PM_{1.0}$ or $PM_{2.5}$ were relatively low throughout the observation period. The $PM_{1.0}$ and $PM_{2.5}$ studied by using the several different instruments have demonstrated that the $PM_{1.0}$ mass concentrations accounted for 70%-80% of the $PM_{2.5}$ concentrations for various non-refractory species.

**4   Conclusions**

This study evaluated the performance of the newly developed Time of Flight-Aerosol Chemical Species Monitor-Capture Vaporizer (TOF-ACSM-CV) with a $PM_{2.5}$ aerodynamic lens, in comparison to other offline and online methods. Additionally, we investigated the chemical compositions of $PM_{1.0}$ and $PM_{2.5}$ using two co-located and identically configured TOF-ACSM-CVs. Measurements were carried out during the RITA-2021 campaign with two intensive observation periods in Spring and

Fall at CESAR (the Cabauw Experimental Site for Atmospheric Research) in the Netherlands. $PM_{1.0}$ and $PM_{2.5}$ were also collected on filters for offline analysis. We observed excellent agreement ($R^2$ from 0.87- 0.99) between the online and offline

measurements with the differences of all secondary inorganic aerosols smaller than 10%. This level of accuracy is significantly higher than the nominal specification of ± 30%, indicating the reliability of the ACSM with CV and $PM_{2.5}$ lens in accurately measuring atmospheric aerosols. The integrated volume size distribution obtained from the MPSS showed a strong correlation with the summed volume concentration calculated from ACSM and MAAP measurements (slope = 1.0, $R^2$ = 0.91). The bias among the multiple online measurements (ACSM, MPSS and MARGA) was dependent on RH, which could be due to the different inlet systems (cyclones vs impactors). However, the good agreements (with all $R^2$ > 0.9) enable further quantification of $PM_{1.0}$ and $PM_{2.5}$ mass concentrations with the ACSM. The average mass concentration of non-refractory (NR) compounds was 4.11 ± 3.32 µg m$^{-3}$ for $PM_{1.0}$ and 5.27 ± 3.98 µg m$^{-3}$ for $PM_{2.5}$. The NR-$PM_{1.0}$ fraction accounted for approximately 70% - 80% of the NR-$PM_{2.5}$ mass concentration, with both dominated by organics (>33%), followed by nitrate (>27%), sulphate (~18%) and ammonium (~17%). However, the mass fraction of nitrate tended to increase by 7.2% (from 38.3% to 45.5%) while the OC mass fraction tended to decrease 4.8% (from 19.9% to 15.1%) from the $PM_{1.0}$ to $PM_{2.5}$. This change reveals the size-dependence on chemical composition. In conclusion, the introduction of the CV and $PM_{2.5}$ lens significantly improved the collection and detection efficiency, enabling the TOF-ACSM to measure the $PM_{1.0}$ and $PM_{2.5}$ substance quantitively with good calibration.

**Data availability**

The data involved in this study is part of the Ruisdael Observatory (https://ruisdael-observatory.nl) project and can be accessed at repository under https://doi.org/10.5281/zenodo.7924288 (Liu et al., 2023). The meteorological data is available at the KNMI Data Platform (https://dataplatform.knmi.nl).

**Author contribution**

XL, BH, and UD designed this study. AH, JM, PY, DD, JB, UD and XL implemented the experiment and sample analysis. XL analysed the data and wrote the manuscript. All co-authors proofread and commented on the paper.

**Competing interests**

The authors declare that they have no conflict of interest.

**Acknowledgements**

The Chinese Scholarship Council (No.201906350118) is acknowledged for the financial support for the author X. Liu. In the project we make use of the Ruisdael observatory infrastructure, funded by the Dutch Science foundation NWO (grant number





184.034.015). The authors would like to thank Jan Pieter Lollinga and Anneliese Kasper-Giebl for their support in data collection.

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
