# Peer review of "Measurement report: Evaluation of the TOF-ACSM-CV for PM1,0 and PM2,5 measurements during the RITA-2021 field campaign"

_EGUsphere, 2023_

## Author Comment (AC1)

Response to reviewers for "Measurement report: Evaluation of the TOF-ACSM-CV for $PM_{1.0}$ and $PM_{2.5}$ measurements during the RITA-2021 field campaign" by Xinya Liu, Bas Henzing, Arjan Hensen, Jan Mulder, Peng Yao, Danielle van Dinther, Jerry van Bronckhorst, Rujin Huang, and Ulrike Dusek. (Manuscript ID: EGUSPHERE-2023-967)

We would like to thank the reviewers for their valuable comments on our paper, we believe that the manuscript has been improved significantly due to their suggestions. To facilitate the review process, we have copied the reviewers' comments in black text and renumbered them for easy cross-referencing. Our responses are in standard blue text. We have responded to all the comments made by the reviewers and have revised the manuscript accordingly.

**Reviewer #1**

Liu et al. reported the cross-comparison between the measurement of a flight-aerosol chemical speciation monitor with the capture vaporizer and a PM2.5 aerodynamic lens (TOF-ACSM-CV-PM2.5) and other offline chemical analyses. Their results show great agreement between online and offline measurements, suggesting TOF-ACSM-CV-PM2.5 has good collection efficiency. The experiments were well designed, the manuscript overall is well written, and the conclusions are well supported. I just have a few minor comments for the authors. Thus, I suggest a minor revision of this manuscript.

Comments:

C1: L115-117, "The mass … (6.6 m2/g)." I think 6.6 m2/g is a mass absorption cross section (MAC), not mass scattering cross section (MSC). Please confirm that and make changes accordingly.

Thanks for the correction. It has been corrected to "mass absorption cross section (MAC)" in the revised manuscript.

C2: Section 2.32 has some issues:

C2.1: Did you pre-bake the quartz filter? Moreover, quartz filters can adsorb gas species. Could you discuss how that could affect your results? I assume that will introduce more OC due to adsorbed VOCs and SVOCs.

Yes, we pre-baked the quartz filters to minimize potential artifacts and used field blanks for correcting all adsorption artifacts during filter handling, storage and analysis.

We agree that the quartz filters have the potential to adsorb organic gas species during sampling. Backup filters can be used as a rough correction for this positive artifact. However, they were not employed in this campaign. Some tests with filters from previous campaigns with a high-volume filter sampler showed that OC on the backup filter was roughly 10-20% of OC on the front filter (Dusek, unpublished data). Even though quartz backup filters only provide a very rough estimate of the positive artifact and a different filter sampler was employed in this study, we estimate that OC on the filters was likely overestimated on the order of up to 20%. We briefly discuss this in the revised manuscript as an additional potential discrepancy between ACSM Organics and filter OC (L245-256). A much higher positive artifact would lead to unrealistically high OM/OC values.

C2.2: Did you weight filters before analysis to see if there was any change in weight during storage?

We did not weigh the quartz filters before analysis to monitor changes in weight during storage. We recognize that monitoring weight changes can be a valuable quality control measure. However, in our case this was not possible as the OC/EC analysis was carried out in a different lab that did not have a filter weighing facility. In general, we believe it is unlikely that the filters lost/gained a lot of material during storage for the following reasons.

The OC/EC analysis was conducted shortly after the campaign, resulting in a very brief storage period of approximately 2 weeks. In the meantime, filters were stored in the freezer (see response for C2.3). Based on our experience and existing literature, short-term storage of quartz filters under these conditions is unlikely to lead to significant weight changes.

In addition, by not doing a weighing we reduce handling of the filters to minimize the risk of contamination, which could potentially impact the results more than the minimal weight changes we anticipated.

C2.3: I would never store filter samples under room conditions since samples might evaporate and/or react during storage. A recently published paper shows that storing filters under room temperature can significantly change the OC due to evaporation and oxidation (Resch et al., 2023). Please add a discussion regarding to potential change of sample due to storage. This is related to your discussion of NO3 loss from the filter in section 3.1.1 since your storage temperature is 20 C. It can also explain why you see more OM from ACSM than filters.

Sorry for the incorrect description in the original text, and it has now been corrected in L165-167. The actual situation is that all the filter samples were stored at -20 ± 0.5 °C and then protected by the ice packs during the transportation. The filter weighing was performed at a temperature of 20.0 ± 0.5 °C and relative humidity of 50 ± 2%. We fully agree with the reviewer's comment that higher storage temperatures lead to evaporation and oxidation of OC and also affect $NO_3$.

C2.4: It is not clear which filter you used for which offline analysis.

Sorry for that, we have clarified this question in the revision version (L165 and L170). The PTFE filters were used for weighing the gravimetric mass, while the quartz filters were used for the ion analysis by the Ion chromatography and EC/OC analysis by the Sunset thermal-optical analyzer.

C2.5: For the thermal-optical method, it distinguishes OC and EC based on the different temperature steps. All CO2 that comes from He/O2 steps are considered to have originated from EC. Moreover, OE-EC has few artifacts due to VOC and SVOCs in the sample (Y. Cheng et al., 2010; Z. Cheng et al., 2019; Turpin et al., 1994, 2000). Also, EC is typically overestimated due to the pyrolysis of OC and strongly absorbing BrC (Z. Cheng et al., 2019). Please add relevant discussion about these artifacts.

Yes, the thermal-optical method to distinguish the OC and EC has various pitfalls, which we now discuss at two points in the manuscript (L248-256 and L280-283). The artifacts regarding EC quantification are not completely understood, and various studies give contradictory results. According to our own studies at the same measurement location, it is actually likely that EC is underestimated by the EUSAAR protocol, especially for more highly loaded filters (Zenker et al., 2020). We included a brief discussion in the main text, where we say that previous studies show evidence for both potential over- or underestimation of EC during thermal-optical analysis. As explained above (C2.1), we roughly estimate the positive OC artifact in our filters in the 20% range.

C3: Figure 1: The caption is not very clear. It should be stated that these dots are daily averages in the caption. What is the variance of online measurements?

Thank you for this suggestion. We have now clarified they are daily average data in the caption of Figure 1.

Regarding to the variability of online measurements (10 mins resolution) during the day, please refer to the figure below. Larger variances are observed in the measurement of $PM_{2.5}$ compared to $PM_{1.0}$, which is expected. Furthermore, for different chemical species, larger variances are observed in $NO_3$, followed by $NH_4$ and $SO_4$, reflecting the variations in the formation of secondary inorganic aerosols during the day. The relevant information has been added in the revised manuscript.

[Figure]

Figure 1: Comparison of the online (ACSM and MAAP) and offline (Filters) daily average mass concentrations of various chemical components. $PM_{1.0}$ is indicated in green and $PM_{2.5}$ in orange. The shaded area represents the 95% confidential interval of the best fit line. Error bars on y axis represent the standard deviation of the measurements during the day.

C4: L198-201, "A well-known … Pakkanen and Hillamo, 2002)." If this is true, you should always see filter has lower NO3 than ACSM. It actually suggests the difference between filter and ACSM should even be larger.

Thanks for the comments. We reformulated the paragraph (L203-204) so that it becomes clear from the start that the evaporation of ammonium nitrate from the filters likely does not play a big role, which is consistent with moderate temperatures and high average RH during our campaign.

C5: L257-263, "The eBC was … a reasonable result." It is also due to MAAP measures eBC, which includes BC and any other species that absorb light at 637 nm (e.g., BrC). The MAAP also overestimates eBC due to multi-scattering effects and loading effects.

Yes, we agree the statement and added the relevant information in the revised manuscript (L274-279). In principle, the MAAP includes corrections for loading effects and multiple scattering (Petzold et al., 2005; Petzold and Schönlinner, 2004), but these corrections are not perfect, as loading and multiple scattering effects may vary depending on specific environmental conditions or instrument sensitivities. Nonetheless, this is likely a second order effect Therefore, we will refrain from an in-depth discussion of this particular issue and mention it only briefly.

**Reviewer #2**

Liu et al. compared the performance of online TOF-ACSM-CV measurements against offline filter sample analysis. The results show that ACSM has a difference less than 10% from filter analysis for all the inorganic species, which indicates the good collection efficiency of the capture vaporizer. Additionally, TOF-ACSM-CV measurements overall agrees with MARGA in terms of inorganic species and with MPSS in terms of particle volume. The reasons for the differences are analyzed and discussed. I find this manuscript easy to read and overall well written. The manuscript can be published after the following minor comments are addressed.

Comments:

C1: Line 112: Do you mean electrostatic losses? Why would stainless steel tubing reduce diffusional losses compared to other tubings?

Thank you for your question. Our original statement was intended to highlight the measures taken to minimize losses in our sampling system. To clarify, stainless steel tubing was primarily chosen to reduce electrostatic losses, not only diffusion losses. The manuscript has been revised according in L112.

C2: Line 165: Why are the filters stored at 20C instead of at low temperatures? How long is it between filter collection and subsequent analysis? Any estimation about OC losses? A discussion on this should be added to section 3.1.1

We apologize for the incorrect formulation in our initial manuscript. To clarify, the filters used in our study were stored at a temperature of $-20 \pm 0.5$ °C. They were conditioned for weighing at 20°C and 50% rh, but this refers only to the Teflon filters. We have now accurately updated this information in L165-167 of the revised paper.

The filters were carefully transported under strict temperature control, using ice packs to maintain the necessary low temperature conditions. We analyzed the filters within two weeks after the campaign. As mentioned in our response to the first reviewer's comments (C2.1 and C2.2), we took measures to avoid the evaporation or absorption of the VOCs and SVOCs on the filters. A brief discussion on its impact on OC estimation has been added to the revised paper L248-256.

C3: Line 321: The newest WHO PM2.5 annual limit is 5ug/m3.

Thank you for highlighting the latest WHO guideline for $PM_{2.5}$, which sets the annual limit at 5µg/m³. We have promptly updated our paper to reflect this new standard. The corresponding sections of the manuscript have been revised to ensure our analysis and discussions are aligned with the latest WHO guidelines.

C4: Line 338: Why does a low fraction of OC in PM 2.5 indicate organic compounds are more abundant in the PM1.0-2.5 range? There is also a typo: similar in PM1.0 and PM2.5

Thank you for your question and correction. We have updated this information in the revised paper L360-362. Given that the mass fraction of Organic Aerosols (OA) is rather consistent in both $PM_{1.0}$ and $PM_{2.5}$, while the proportion of OC is higher in $PM_{1.0}$, it may imply a predominance of pure carbonaceous compounds (such as hydrocarbons) in the $PM_{1.0}$. In our study, the Organic Matter (OM) to Organic Carbon (OC) ratios were $1.58 \pm 0.54$ for $PM_{1.0}$ and $1.97 \pm 0.59$ for $PM_{2.5}$. The lower OM/OC ratio in $PM_{1.0}$ points to a predominance of hydrocarbon-like aerosols in smaller particles, whereas the higher ratio in $PM_{2.5}$ suggests a greater presence of more oxidized aerosols in larger particles.

C5: Figure S1: it should show chlorine and OA, not EC and OC.

Thank you for pointing out this mistake. Figure S1 has been corrected in the revised version.

C6: Do the authors suggest an aerosol drying protocol to decrease the effect of humidity on the cut-off size?

Based on the results of our study it would definitely beneficial to select the particle cut-off sizes at a defined humidity rather than the varying ambient humidity. This humidity should preferably not be so low to dry the particles out completely to prevent particle bounce in inlet impactors. However, this is practically not very easy to achieve, so we are not optimistic that this will become a standard operating procedure for stations and field campaigns very soon.

**References**

Petzold, A. and Schönlinner, M.: Multi-angle absorption photometry—a new method for the measurement of aerosol light absorption and atmospheric black carbon, J. Aerosol Sci., 35, 421–441, https://doi.org/https://doi.org/10.1016/j.jaerosci.2003.09.005, 2004.

Petzold, A., Schloesser, H., Sheridan, P. J., Arnott, W. P., Ogren, J. A., and Virkkula, A.: Evaluation of multiangle absorption photometry for measuring aerosol light absorption, Aerosol Sci. Technol., 39, 40–51, https://doi.org/10.1080/027868290901945, 2005.

Zenker, K., Sirignano, C., Riccio, A., Chianese, E., Calfapietra, C., Prati, M. V., Masalaite, A., Remeikis, V., Mook, E., Meijer, H. A. J., and Dusek, U.: δ13C signatures of organic aerosols: Measurement method evaluation and application in a source study, J. Aerosol Sci., 145, 105534, https://doi.org/10.1016/j.jaerosci.2020.105534, 2020.

---

## Editor Decision (ED1)

I would like to thank the authors for promptly replying to all the reviewers' comments and corrections. I have only a few minor points to be further clarified. After that, I believe the manuscript will be accepted. Thus, please consider the points listed below.

1 – Could you please add a sentence to the manuscript to clarify about the pre-baking of the quartz filters?

2 – P10, L248: "indicated by a slope of from $2.21 \pm 1.27$ for PM1.0 and $2.27 \pm 0.92$ for PM2.5", are the slope values correct? I mean, if they are related to Fig. 1, then the slopes indicated are different (2.77 and 2.11). Could you please verify?

3 – In the answer to the Reviewer #1 (C2.1) the authors estimate the positive artefacts, based on previous unpublished data, up to 20%. However, in the manuscript (P.10, L256) the authors mention 20-30%. Could you please clarify which one is correct? Or clarify if they refer to different artefacts.

---

## Author Response (AR2)

Response to editor for "Measurement report: Evaluation of the TOF-ACSM-CV for $PM_{1.0}$ and $PM_{2.5}$ measurements during the RITA-2021 field campaign" by Xinya Liu, Bas Henzing, Arjan Hensen, Jan Mulder, Peng Yao, Danielle van Dinther, Jerry van Bronckhorst, Rujin Huang, and Ulrike Dusek. (Manuscript ID: EGUSPHERE-2023-967)

Thank you to the editor for the thoughtful comments. We have considered and addressed each point in the following sections, and have made the necessary updates to the manuscript accordingly.

Editor comments:

I would like to thank the authors for promptly replying to all the reviewers' comments and corrections. I have only a few minor points to be further clarified. After that, I believe the manuscript will be accepted. Thus, please consider the points listed below.

1 – Could you please add a sentence to the manuscript to clarify about the pre-baking of the quartz filters?

In the revised manuscript, we have included an additional sentence at Line 165 to clarify the procedure for pre-baking the quartz filters

2 – P10, L248: "indicated by a slope of from 2.21 ± 1.27 for PM1.0 and 2.27 ± 0.92 for PM2.5", are the slope values correct? I mean, if they are related to Fig. 1, then the slopes indicated are different (2.77 and 2.11). Could you please verify?

We are sorry for the typo mistake and have corrected it in the revised manuscript.

3 – In the answer to the Reviewer #1 (C2.1) the authors estimate the positive artefacts, based on previous unpublished data, up to 20%. However, in the manuscript (P.10, L256) the authors mention 20-30%. Could you please clarify which one is correct? Or clarify if they refer to different artefacts.

Thank you for your careful observation. Based on our previous campaign data, high-volume filter samplers indicated that organic carbon (OC) on the backup filter was approximately 10-20% of that on the front filter (Dusek, unpublished data). However, considering that a different sampler was employed in the current study's campaign, we conservatively estimated that this discrepancy could be between 20-30% as we indicated in the manuscript.

Remarks from the preceding review file validation

We have improved Figures 2, S5, and S6 to ensure accessibility for readers with color vision deficiencies.